# Small-Cell Lung Cancer: Is Liquid Biopsy a New Tool Able to Predict the Efficacy of Immunotherapy?

**DOI:** 10.3390/biom14040396

**Published:** 2024-03-25

**Authors:** Rossella Fasano, Simona Serratì, Tania Rafaschieri, Vito Longo, Roberta Di Fonte, Letizia Porcelli, Amalia Azzariti

**Affiliations:** 1Laboratory of Experimental Pharmacology, IRCCS Istituto Tumori Giovanni Paolo II, V.Le O. Flacco, 65, 70124 Bari, Italy; r.fasano@oncologico.bari.it (R.F.); t.rafaschieri@oncologico.bari.it (T.R.); r.difonte@oncologico.bari.it (R.D.F.); l.porcelli@oncologico.bari.it (L.P.); a.azzariti@oncologico.bari.it (A.A.); 2Medical Thoracic Oncology Unit, IRCCS Istituto Tumori Giovanni Paolo II, 70124 Bari, Italy; v.longo@oncologico.bari.it

**Keywords:** small-cell lung cancer (SCLC), immunotherapy, circulating tumor cells (CTCs), circulating tumor DNA (ctDNA), soluble factors, extracellular vesicles (EVs)

## Abstract

Small-cell lung cancer (SCLC) cases represent approximately 15% of all lung cancer cases, remaining a recalcitrant malignancy with poor survival and few treatment options. In the last few years, the addition of immunotherapy to chemotherapy improved clinical outcomes compared to chemotherapy alone, resulting in the current standard of care for SCLC. However, the advantage of immunotherapy only applies to a few SCLC patients, and predictive biomarkers selection are lacking for SCLC. In particular, due to some features of SCLC, such as high heterogeneity, elevated cell plasticity, and low-quality tissue samples, SCLC biopsies cannot be used as biomarkers. Therefore, the characterization of the tumor and, subsequently, the selection of an appropriate therapeutic combination may benefit greatly from liquid biopsy. Soluble factors, circulating tumor DNA (ctDNA), circulating tumor cells (CTCs), and extracellular vesicles (EVs) are now useful tools in the characterization of SCLC. This review summarizes the most recent data on biomarkers detectable with liquid biopsy, emphasizing their role in supporting tumor detection and their potential role in SCLC treatment choice.

## 1. Introduction

Lung cancer is the most common cancer in males and the third most common cancer in females, meaning it is the number one cause for cancer-related death in males and number two for females. About 250,000 patients are diagnosed with SCLC each year globally, of which, approximately 200,000 succumb to the disease [1]. SCLC is characterized by a rapid doubling time and the early development of widespread metastases [2]. Smoking is known to be the primary risk factor for the development of lung cancer, and it accounts for more than 95% of SCLC cases [1]. At diagnosis, 60–65% of SCLC patients present metastatic disease with the rapid onset of symptoms, including dyspnea, chest pain, cough, and weight loss. Due to a high percentage of brain metastases, neurologic symptoms are also frequent at diagnosis [3]. SCLC is typically divided into two stages based on the extent of the disease, namely limited-stage (LS)-SCLC and extensive-stage (ES)-SCLC [3]. However, the prognosis is unfavorable in both forms, with a median survival (SM) ranging from 15 to 20 months for the limited form and 8 to 13 months for the diffuse form [4]. Concerning ES-SCLC, platinum-based chemotherapy (CT) has been the first-line standard of care for almost 40 years, generally combined with etoposide or irinotecan [2].

In the last few years, the addition of immune checkpoint inhibitors (ICIs) to platinum-based CT has improved the outcome of ES-SCLC patients in the first-line setting (Table 1). The phase III IMpower133 trial firstly described a statistically significant overall survival (OS) benefit by adding atezolizumab, an anti-PD-L1, to carboplatin plus etoposide CT [3,5]. Subsequently, the phase III Caspian trial, evaluating first-line anti-PD-L1 durvalumab in combination with etoposide plus either cisplatin or carboplatin, confirmed a significant improvement in OS [6,7]. Both these ICIs have been approved in clinical practice. Recently, two other phase III trials, namely CAPSTONE-1 and ASTRUM005, confirmed the benefit of immunotherapy plus CT in the first-line treatment of ES-SCLC patients by adding the anti-PD-L1 adebrelimab [8] and the anti-PD1 serplulimab [9], respectively. Moreover, in the WCLC 2023, the results of the phase III RATIONALE-312 trial showed OS and overall response rate benefits by adding tislelizumab, an anti-PD1, to first-line CT [10]. Finally, a novel PD-L1 inhibitor, benmelstobart, and the antiangiogenic anlotinib plus CT showed a historically long OS of 19.32 months [11]. Nevertheless, some ES-SCLC patients benefit from ICIs, but differently to non-SCLC (NSCLC), the biomarker selection is lacking.

SCLC has a lower PD-L1 expression (15%) than NSCLC (60%), since PD-L1 is mainly expressed on immune cells compared to tumor cells, and it displays an immunological microenvironment that is depleted during tumor infiltration [12,13,14]. Unfortunately, the scarce cellularity of SCLC specimens limits the ability to detect PD-L1 [15]. For example, only a third of patients enrolled in the IMPower133 trial had evaluable tumor tissue, with negative PD-L1 tumor cell expression in almost all cases, while the PD-L1 expression in immune cells was negative in about half of the cases, without any correlations between PD-L1 expression levels and clinical outcomes [3]. Similarly, the analysis of the phase III CASPIAN trial failed to demonstrate any correlation between PD-L1 expression and treatment efficacy [16]. Despite SCLC being characterized by a high tumor mutational burden (TMB), due to the association of SCLC with smoking, it is not suitable as a predictive biomarker for ES-SCLC patients receiving first-line chemoimmunotherapy [3,17].

In recent years, several studies have been exploring a theoretical subtype subdivision of SCLC. In particular, Gay et al. [18] identified four SCLC subtypes, by the differential expression of transcription factors, as ASCL1, NEUROD1, and POU2F3, and, by the low expression of all three transcription factors with an inflamed gene signature, as SCLC-I. This latter subtype is characterized by the higher expression of CD8+ T cells, NK cells, macrophages, and B-lymphocytes. Moreover, SCLC-I expressed higher levels of HLAs and immune checkpoints including PD-L1, PD1, CTLA4, TIGIT, LAG3, IDO1, and CD38. Subtype-by-subtype analysis comparing survival between chemotherapy plus immunotherapy and chemotherapy alone arms showed a higher survival benefit from the addition of immunotherapy in SCLC-I [18,19]. However, SCLC is a heterogeneous disease; a single patient presents several subtypes. Moreover, SCLC is also characterized by high levels of tumor plasticity, with SCLC cells being able to switch from one subtype to another [18]. Considering the scarce cellularity of SCLC specimens and the high heterogeneity and plasticity of SCLC, liquid biopsy may become an excellent tool in the characterization of tumors and in the search for an adequate therapeutic approach [20]. In lung cancers, several aspects are being studied, including circulating proteins, circulating tumor DNA (ctDNA), circulating tumor cells (CTCs) and extracellular vesicles (EVs), which are now useful tools in the discrimination of histological subtypes [21] and in the detection of specific mutations [22].

In particular, in SCLC, liquid biopsy could be more useful than tumor tissue specimens due to the difficult acquisition of tumor biopsies after relapse and the intra-/inter-tumoral heterogeneity of SCLC, which is better detectable in serial specimens [23].

## 2. Circulating Tumor Cells (CTCs)

Primary tumors release circulating tumor cells (CTCs) into the circulation, which migrate and spread to secondary sites to establish metastasis. These cells are found in limited numbers in most advanced cancers; however, SCLC is an exception due to the high number of CTCs released into circulation. The average count is 400 CTC/7.5 mL of blood that can reach peaks of several thousand cells/7.5 mL of blood in advanced metastatic stages [24,25]. In general, CTCs represent a very heterogeneous group for the expression of surface markers or physical characteristics, such as cell size, deformability, electrical charge, etc. [26].

CTCs have now been shown to be a prognostic indicator in patients with SCLC (Table 2). Patients with elevated CTC levels before chemotherapy or radiochemotherapy had worse prognoses [27,28,29] and patients with a CTC count of less than 12 after chemotherapy had a significantly lower risk of progression and death [30].

Metastasis involves several crucial steps such as intravasation, angiogenesis, and extravasation, all mechanisms related to SCLC progression that have been abundantly studied but still not fully resolved. Epithelial–mesenchymal transition (EMT) has a major role in metastasis development [31].

It is known that in the delicate process of intravasation, some of the surviving cells may arrest in the vascular lumen and extravasate through the endothelium, reaching the parenchyma of distant organs. In the new stromal environment, a small group of tumor cells forms micrometastases [32]. As the metastasis again shows an epithelial phenotype, the migratory and invasive cells need to reverse EMT with a loss of these migratory and invasive properties; they need to undergo apico-basal polarization and begin to express junctional complexes, creating the so-called mesenchymal–epithelial transition (MET) [33]. However, these transitions are not necessarily so sharp; cancer cells with hybrid phenotypes consisting of epithelial and mesenchymal features have been observed [34]. There is conflicting information about EMT, which is thought to aid in the spread of tumor cells. It has been shown that some active compounds derived from tobacco combustion can induce EMT through hypoxia, inflammation, and oxidative damage, which are biological processes triggered by cigarette smoke [35]. Another significant factor is that all SCLC cells have p53 mutations, and p53 loss decreases miR-200c, promoting EMT [36]. Furthermore, Hamilton G. states that in SCLC, local inflammatory conditions at the primary tumor site and the recruitment of tumor-associated macrophages (TAMs) appear to increase the spread of tumor cells into the circulation in processes that may proceed independently of EMT. In particular, SCLC CTCs recruit TAMs and stimulate the secretion of chitinase-3-like-1 (CHI3L1), vascular endothelial growth factor (VEGF), and matrix metalloproteinase-9 (MMP9), enabling SCLC CTCs to disseminate bypassing the mesenchymal phase [37]. Moreover, SCLC CTCs express high levels of EpCAM and E cadherin and shape organized tumorospheres with close cell–cell contacts, suggesting a direct dissemination of epithelial-like tumor cells without the need for an EMT-MET cycle [38]. Conversely, TAMs may promote EMT by IL-8 [39], and SCLC CTC cell lines express variable amounts of vimentin. However, it is unclear whether the expression of mesenchymal markers in these cells indicates a previous EMT or the consequence of deregulated gene expression in SCLC cells [33].

Vascular mimicry (VM), the ability of tumor cells to imitate the development of endothelial-like vessels, is a crucial step in the development of metastases [40]. Notably, VM-forming tumor cells have highly upregulated EMT regulators and EMT-related transcription factors, suggesting that EMT may be essential for VM formation. Thus, other factors as well as the upregulation of EMT-associated adhesion molecules may contribute to the process of VM formation [41]. Williamson and colleagues demonstrated the existence of rare subpopulations of CTCs co-expressing epithelial cadherin (E-cadherin) and some cytokines typical of VM in patients with SCLC [42], thus stating that higher levels of VM are associated with worse overall survival in patients with limited-stage SCLC (*p* < 0.025), an event likely due to the increased access of CTCs to the circulation [42].

Many immune cells are drawn into the tumor microenvironment (TME) during tumorigenesis to either directly or indirectly contribute to the development of the tumor by secreting growth factors and chemokines, which are highly correlated with the formation of VM. Additionally, they can affect the progression of the tumor by controlling the formation of VM [43]. CTCs can enter the circulation through irregular tumor vessels as single cells or circulating tumor micro-emboli [44]. Certainly, the small cell size (about 8 μm) and high pulmonary blood perfusion are crucial factors for the spread of CTCs [45,46]. It has been shown that although clusters of CTCs are rare in peripheral blood, they have greater metastatic potential than single CTCs [47] and that, in contrast, patients who had no clusters at baseline had significantly better progression-free survival (PFS) (8.2 vs. 4.6 months, *p* < 0.001) and OS (10.4 vs. 4.3 months, *p* < 0.001) than patients who had clusters of CTCs at the same time-point [44]. It has also been shown that patients who have clusters after the completion of the chemotherapy cycle have a higher risk of relapse [30]. In the case of CTCs in the form of single cells, although they are chemosensitive, tumors that have relapsed after the first treatment show general drug resistance, so the reduction in CTCs does not appear to correlate with the response of the resident lesions [48]. Hou et al. found that patients with SCLC and a CTC count ≥ 50 CTC/7.5 mL of blood before chemotherapy have a worse clinical outcome in terms of OS than those with a CTC count < 50 CTC/7.5 mL of blood (HR = 2.5) [44]. However, precise indications for diagnostic or therapeutic purposes regarding CTC count are not yet defined [49].

While the immune system plays a significant role in several CTC-related processes, including VM and EMT, not enough is known about the precise role that CTCs may play in the response to immunotherapy in SCLC. Regarding SCLC, cell lines stabilized by CTCs show the spontaneous formation of large spheroids, called tumorspheres, which have a rim of proliferating cells and a hypoxic core with necrotic cells and are formations that show markedly high resistance to chemotherapeutics compared with the same cells in single-cell suspensions [48]. Hamilton et al. observed that some SCLC CTC primary cell lines lack PD-L1 expression and may protect cells from immune attack due to their high resistance to anticancer drugs [50]. Overall, research on CTCs is still limited today because of the scarcity of short- or long-term cultured samples and the heterogeneity of these cells that make them unrepresentative of the tumor mass.

**Table 2 biomolecules-14-00396-t002:** CTCs as potential biomarkers. ES: extensive-stage (ES)-SCLC, LS: limited-stage (LS)-SCLC.

Potential Biomarkers	Stage	Summary	References
CTCs			
CTCs	ES	A blood sample containing 400 CTCs has an average count of 7.5 mL of blood, but in advanced metastatic stages, this can peak at several thousand cells per 7.5 mL of blood.	[24,25]
CTCs	ES, LS	Prior to chemotherapy or radiochemotherapy, patients with higher CTC levels had a worse prognosis.	[27,28,29]
CTCs	/	SCLC CTCs recruit macrophages; elicit the secretion of various cytokines; and express CHI3L1, VEGF and MMP9, developing the abilities of aggressiveness and invasion.	[37]
CTCs	/	High levels of EpCAM and E cadherin are expressed by SCLC CTC lines, indicating direct dissemination without the need for an EMT-MET cycle.	[38]
Rare CTC subpopulations	LS	Rare subpopulations of CTCs express vasculogenic mimicry (VM) markers. Patients with high levels of VM have shorter OS.	[42]
CTCs	/	Before treatment, patients with a CTC count ≥ 50 CTC/7.5 mL of blood had a worse OS compared to those whose CTC count s < 50 CTC/7.5 mL of blood.	[44]
CTCs	/	High pulmonary blood perfusion and small cell size (around 8 μm) are crucial for the dissemination of CTCs.	[45]

## 3. Circulating Tumor DNA (ctDNA)

Circulating tumor DNA (ctDNA) holds great promise as a non-invasive biomarker for profiling cancer mutations [51]. ctDNA is fragmented DNA originating from tumors that circulates in the bloodstream alongside cell-free DNA (cfDNA) from various other sources. While the precise mechanisms responsible for the release of ctDNA into circulation have not been completely elucidated, most studies suggest that the apoptosis and/or necrosis of tumor cells are the primary sources of ctDNA.

Only a portion of the DNA present in blood plasma originates from cancer cells [52]. In metastatic diseases, ctDNA (circulating tumor DNA) makes up a larger proportion of cfDNA (cell-free DNA) compared to earlier stages of cancer or normal tissues. This increase in ctDNA is likely since cancer cells in advanced tumors release more genetic material into the bloodstream due to their rapid growth and high cell turnover. Patients with a high tumor burden (i.e., a large amount of cancer tissue) and aggressive disease tend to have even higher proportions of ctDNA in their cfDNA. In some cases, ctDNA can make up more than 90% of the cfDNA present in the bloodstream of these patients. This high content of ctDNA in the bloodstream can serve as a negative prognostic indicator [53].

Extensive research has established that circulating DNA can indeed carry oncogenes. Clinical data further demonstrate that these oncogenes, as well as other tumor-specific elements like hypermethylated tumor suppressor genes, abnormal microsatellites, and rearranged chromosomes, can be detected in blood plasma and are identical to those found in the corresponding tumor tissue within the same patient. This consistency between blood plasma and tumor tissue markers underscores the diagnostic and prognostic potential of liquid biopsy approaches for cancer detection and monitoring.

The results of in vitro experiments indeed suggest that cell-free DNA can enter and transform normal cells into neoplasms. This implies that circulating DNA containing oncogenes could function somewhat like an intrinsic oncovirus. The transfection of internal cf-DNA could serve as an alternative pathway for the initiation of cancer metastasis. Further research in this area could lead to important insights into cancer prevention and treatment [54].

Advances in the field of DNA methylation research have created new opportunities for addressing difficulties in the analysis of tissue biopsies from SCLC patients.

Chemi et al. investigated the DNA methylation profiling of both patient-derived models and circulating cell-free DNA (cfDNA). Tumor-specific methylation patterns were detected in cfDNA samples from SCLC patients and correlated with survival outcomes [55]. The significance of this analysis was confirmed in a recent study conducted on two cohorts of 179 patients, where DNA methylation patterns specific to different SCLC subtypes were detected in both the tumor tissue and blood of SCLC patients [56]. Both studies highlight the clinical utility of cfDNA methylation profiling as a liquid biopsy approach applicable for detecting information on the biological behavior and clinical course of different subtypes, including dynamic changes with disease progression.

Thus, the presence and proportion of ctDNA in the bloodstream can provide valuable insights into the aggressiveness and progression of cancer, especially in patients with advanced and metastatic disease. Monitoring ctDNA levels can be a valuable tool in assessing prognosis and guiding treatment decisions.

The levels of ctDNA in SCLC patients exhibited dynamic fluctuations throughout the course of the disease. Initially, there was a reduction in ctDNA levels following the completion of two rounds of chemotherapy, likely due to the sensitivity of SCLC to this treatment. Consequently, the tumor burden diminished after chemotherapy, but as the disease advanced, there was a subsequent increase in ctDNA levels [57].

Furthermore, ctDNA is a predictor of outcome in SCLC patients receiving atezolizumab, and it has a strong correlation with prognosis in SCLC patients receiving second-line immunotherapy [58]. We eagerly await the results of several investigations, especially prospective ones looking into the role of ctDNA in predicting the efficacy of immunotherapy in SCLC, given that patterns of ctDNA levels change in parallel with disease progression, suggesting that ctDNA detection holds promise as an effective biomarker for the real-time monitoring of changes during SCLC treatment (Table 3).

## 4. Soluble Factors

Research on soluble factors represents a promising direction for the identification of patients who may benefit from immunotherapy. In the last few years, an increase in soluble ICAM-1 (sICAM-1) expression has been highlighted in several solid tumors, such as gastrointestinal cancer, melanoma, breast cancer, and non-small-cell lung cancer (NSCLC), which is correlated with advanced disease stage and aggressive tumor behavior [59]. Kotteas et al. investigated sICAM-1 expression in 50 SCLC patients at baseline compared with 27 healthy smokers, demonstrating a significantly higher level in patients with respect to healthy subjects, which is directly related to disease progression; these data were consistent with a 9% elevated risk of death for patients with 10 more sICAM-1 units than others [59]. Additionally, they observed a reduction of 25.8% in sICAM-1 level during chemotherapy in SCLC patients with extensive disease [59].

Similarly, a significantly higher level of soluble DR5 (sDR5) was found, which is a factor involved in apoptosis signaling, in SCLC patients compared to healthy controls. Moreover, Wan et al. demonstrated decreased sDR5 levels in the sera of patients after treatment with first-line chemotherapy which correlated with response to treatment (*p* < 0.0001); however, there was no significant correlation between the pre-treatment level of sDR5 and treatment response (*p* < 0.62) [60].

To date, no investigation has been carried out to evaluate sICAM-1 and sDR5 levels after immunotherapy.

PD-L1, expressed by tumor and immune cells in a membrane-bound form, is a biomarker of poor prognosis in some solid tumors including NSCLC. It also exists in a soluble form, sPD-L1, which is involved in immunosuppression and resistance to ICI therapy, with higher concentrations in cancer patients compared to healthy subjects [61]. It is known that high levels of sPD-L1 in the serum of SCLC patients without other immune-related diseases or inflammation reflect the extent of the immune suppression induced by PD-1 and PD-L1 [62]. Accordingly, a 40% increased risk of death was demonstrated in this subset of SCLC patients, thus suggesting the prognostic significance of serum PD-L1 in chemotherapy response [62].

However, So Yeon Oh et al. analyzed serum PD-L1 levels in patients with various solid tumors treated with ICI (ipilimumab), and they observed no relevant differences in soluble PD-L1 levels before and after immunotherapy in SCLC patients, thus excluding PD-L1 as a predictive biomarker of response to therapy with ipilimumab [61].

Other soluble biomarkers involved in immune response signaling include cytokines. A study on two cohorts of SCLC patients treated with chemotherapy alone (*n* = 47) and SCLC patients treated with chemotherapy plus ipilimumab (*n* = 37) assessed the benefits derived from ipilimumab addition by showing different effects of ICI in soluble cytokine levels [63]. They observed that ipilimumab induced a global increase in cytokine concentration, released by both cancer and immune cells, counteracting the effects of chemotherapy. In detail, high baseline levels of IL-2, which promote the initiation and maintenance of immune response, characterized patients with longer OS, acting as a predictive factor of response to ipilimumab treatment. Instead, since TNFα and IL-6 showed a direct correlation with shorter OS, they could be defined as factors predicting resistance to immunotherapy with ipilimumab [63]. Lastly, high levels of IL-4 in patients treated with chemotherapy alone were associated with worse OS, while the elevated concentration of the same cytokine in patients treated with ipilimumab characterized a better outcome [63] (Table 4).

## 5. Extracellular Vesicles (EVs) in SCLC

In recent years, EVs have entered the scenario of cellular components, with a crucial role in the regulation of intercellular communication associated with pathological processes, such as cancer, other than physiological ones. Their structure mimics the cell of origin as they are delimited by cell membranes and consist of proteins, lipids, and nucleic acids similar to the origin cells [64]. They are classified based on their function or size but, given the difficulty of their characterization, the guidelines of the International Society for Extracellular Vesicles (ISEV) suggest utilizing extracellular vesicles for all of them [65].

EVs’ clinical potential is due to their connection with the biological processes that are affected by them. In detail, EVs can be considered as biomarkers for liquid biopsy, aiding in the diagnosis of cancers, such as melanoma, etc.; in the determination of progression in colorectal cancer (CRC), hepatocellular carcinoma, NSCLC, and ovarian tumors; and in the prediction of responses to anti-tumor therapies such as immunotherapy [66,67,68,69,70,71].

Recently, we demonstrated how these nanovesicles can also be useful for monitoring immunotherapy in melanoma by revealing the appearance of induced resistance to anti-PD1 [72]. Other roles of EVs are closely connected to their cargo; in fact, these, by transporting miRNAs, can influence the cellular processes of target cells [70,73] or can themselves be nanodevices for the transport of drugs [74]. Clinical treatment failure of SCLC is a hot topic, and the investigation of mechanisms responsible for the acquisition of resistance to chemotherapy and immune therapy is essential to try to overcome it [7,75]. After surgical resection, the most effective therapy for SCLC is considered DDP-based chemotherapy (cis-diamminedichloroplatinum); however, the resistance, acquired or intrinsic, to this therapeutic approach affects patient response, resulting in a median survival time of only 4–5 months after chemotherapy. Since, in recent decades, SCLC therapies have not greatly improved, other than with platinum compounds, an urgent need in SCLC treatment is to highlight the mechanism responsible for acquired drug resistance, and some authors have demonstrated that EV cargo molecules, such as non-coding RNA (ncRNA) (circular RNA (circRNA) and long non-coding RNA (lncRNA)), are involved in such mechanisms, leading to these subpopulations of EVs being considered as predictors of drug response.

In lung cancer, Rolfo and coauthors were the first to argue that EVs were factors in the liquid biopsy family because it has been possible to determine EGFR mutations and ALK translocations in them. Thus, they suggested EVs as possible biomarkers for the response or resistance to drugs such as osimertinib [76]. From the beginning, another role in lung cancer for EVs has been hypothesized: diagnostic biomarkers. Sandfeld-Paulsen suggested the utilization of exosome-protein profiling as a promising diagnostic tool in lung cancer, even if its utility for SCLC was not validated [77]. A few years later, Wang and coauthors discussed the activity of EVs, originating from malignant pleural effusions, in promoting the invasion and migration of tumor cells due to miR-665, which represses the Notch pathway [78]. Therefore, ultimately, EVs can be, in NSCLC and SCLC, biomarkers of diagnosis and prognosis (Table 5).

Furthermore, Wang and co-authors demonstrated that EVs, released from cancer-associated fibroblasts (CAFs) by the delivery of MEG3 lncRNA, regulate the miR-15a-5p/CCNE1 axis and promote DDP chemoresistance [79]. Conversely, Chao et al. showed that EV-derived circSH3PXD2A inhibits the chemo-resistance and proliferation of SCLC in vitro and in vivo through the miR-375-3p/YAP1 axis, suggesting a role for this circRNA as a predictive biomarker for DDP-resistant SCLC patients [80]. Another circRNA which has recently been related to resistance to platinum compounds in SCLC is hsa_circ_0041150. Zhang and co-authors demonstrated that it promotes both cancer progression and cisplatin resistance in SCLC models, suggesting this circRNA has a role as a biomarker for monitoring chemotherapy resistance in patients treated with cisplatin-based chemotherapy. They showed its higher value as a diagnostic biomarker with respect to traditional cancer-associated markers such as CA125, CA211, and ProGRP in SCLC and its modulation in the function of treatment response, passing from an already higher value present in patients compared to healthy people, which tends to decrease when the patient responds to therapy and then rises again when resistance appears [81].

What is known about the role of EVs in modulating immunotherapy response, the new standard of care for SCLC?

The poor response to anti-PD-1/PD-L1 monotherapy and the well-known characteristic of this tumor of having a highly immunosuppressive tumor microenvironment suggested that the effectiveness of combined chemo-immunotherapy needed to be explored. This strategy led to the recently FDA-approved combination of atezolizumab and durvalumab with platinum–etoposide [82]. Unfortunately, compared with NSCLC, the benefit derived from the addition of immunotherapy to platinum-based CT is lower in SCLC patients, and strategies to improve ICIs’ efficacy and biomarkers of response are urgently needed.

In several tumors, the characterization of the role of EVs in innate and adaptive immunity highlighted that these nanovesicles may have both a pro- and an anti-inflammatory role when they are released from immune cells responsible for the inflammation or impact on T and B cells mainly involved in adaptive immunity. Their pro-inflammatory activity is mediated by the transfer of cytokines, danger signals, and RNAs which are involved in the activation, polarization, and other functions of innate immune cells [83]. Moreover, EVs released from tumor cells and positive for the expression of PD-1 may bind the immune checkpoint inhibitors, such as nivolumab, reducing their pharmacological activity [71].

The cross-talk between EVs and immunity in SCLC has been highlighted by Rao et al., who showed that the polarization of infiltrating macrophages (MØ) toward the M2 phenotype is mediated by tumor-derived exosomes via the NLRP6/NF-κB pathway, promoting SCLC metastasis in vitro and in vivo [84]. The ability of M2-TAM to regulate anti-PD-1/PD-L1 immunotherapy through the release of miRNAs, such as miR-21 and miR-155-5p, is known, which thereby inhibits the T-cell immune response [85].

The relevance of EVs in the response to immunotherapy with anti-PD-L1 in SCLC was explored by Duo and co-authors, who hypothesized that SCLC cells generate EVs which may be mediators of tumor progression and immunosuppression. These EVs, with the PD-L1 expressed on the surface, might bind PD1 on CD8+ T cells, inhibiting their activation [86].

Although studies on the role of EVs in SCLC as a biomarker of the diagnosis, progression, or prediction of response to chemotherapy or immunotherapy are currently scarce, we believe that it could be a hot topic to seek new strategies for the best selection of patients to be subjected to different therapeutic schemes and promising targets in order to hypothesize new therapeutic approaches.

**Table 5 biomolecules-14-00396-t005:** EVs as potential biomarkers.

Potential Biomarkers	Stage	Summary	References
EVs			
Exosomal lncRNA SCIRT/miR-665	/	Plasma exosomal miR-665 and SCIRT are considerably elevated in lung cancer patients and correlate with the disease’s advanced stage, which suggests that the regulatory pathway may have diagnostic and prognostic value.	[78]
Exosomal MEG3 LncRNA	/	Through the MiR-15a-5p/CCNE1 axis, MEG3 lncRNA from exosomes released from CAF increases cisplatin chemoresistance in SCLC.	[79]
Extracellular vesicles derived circSH3PXD2A	/	The EV-derived circSH3PXD2A suppresses SCLC chemoresistance via the miR-375-3p/YAP1 axis. CircSH3PXD2A produced from EVs might act as a prognostic biomarker for DDP-resistant SCLC patients.	[80]
Extracellular vesicles hsa_circ_0041150	/	EVs hsa_circ_0041150 could be a biomarker for monitoring chemotherapy resistance in SCLC patients.	[81]
NLRP6	/	By activating NLRP6, SCLC-derived exosomes cause the immunosuppression of distant macrophages, promoting systemic metastasis.	[84]
Extracellular vesicles containing PD-L1	/	EVs containing PD-L1 might bind PD1 to CD8+ T cells, inhibiting their activation. EVs with PD-L1 had a significant association with progression-free survival.	[86]

## 6. Conclusions

Considering three characteristics of SCLC, high heterogeneity, elevated cell plasticity, and low-quality tissue samples, liquid biopsy may be a useful tool for therapeutic decision making in these patients. The feasibility of liquid biopsy can overcome the limits of SCLC tissue samples. Moreover, serial liquid biopsy permits re-assessment during and after treatment, facilitating changes in therapy before clinical deterioration. Nowadays, liquid biopsy has a major role in SCLC studies, also improving the quantity and quality of information on the development of precision medicine in this recalcitrant tumor.

## Figures and Tables

**Table 1 biomolecules-14-00396-t001:** Phase III studies in ES-SCLC. OS: overall survival, HR: hazard ratio, CT: chemotherapy, ICIs: immune checkpoint inhibitors.

References	Clinical Trial	Drugs	OS (Months)		HR
			CT + ICIs	CT	
Horn L, 2018 [5]	IMpower133	Atezolizumab, carboplatin plus etoposide	12.3	10.3	0.70
Goldman JW, 2021 [6]	CASPIAN	PE plus durvalumab or PE plus durvalumab and the CTLA-4 inhibitor Tremelimumab	12.9	10.5	0.75
Wang, J, 2022 [8]	CAPSTONE-1	Adebrelimab, carboplatin plus etoposide	15.3	12.8	0.72
Cheng, Y, 2022 [9]	ASTRUM005	Serplulimab, carboplatin plus etoposide	15.4	10.9	0.63
Cheng, Y, 2023 [10]	RATIONALE-312	PE plus tislelizumab	15.5	13.5	0.75
Cheng, Y, 2023 [11]	ETER701	Benmelstobart with anlotinib, carboplatin plus etoposide	19.3	11.9	0.61

**Table 3 biomolecules-14-00396-t003:** ctDNA as potential biomarker.

Potential Biomarkers	Stage	Summary	References
ctDNA			
ctDNA	ES	ctDNA levels can represent tumor burden and predict PFS.	[54]
ctDNA	/	Linked to a poor prognosis with a shorter PFS and OS, as well as a lower therapeutic response, particularly in patients receiving anti-PD-L1 immunotherapy.	[57]
ctDNA	/	A predictor of outcome in patients receiving atezolizumab, and it has a strong correlation with prognosis in patients receiving second-line immunotherapy	[58]

**Table 4 biomolecules-14-00396-t004:** Soluble factors as potential biomarkers.

Potential Biomarkers	Stage	Summary	References
Soluble Factors			
sICAM-1	ES	Significantly decreased during chemotherapy.	[59]
sDR5	/	Decreased after first-line chemotherapy and correlated with treatment responses.	[60]
sPD-L1	/	Elevated serum concentration of sPD-L1 might be an independent risk factor for non-response to chemotherapy.	[61,62]
Cytokines	/	High levels of TNF-alpha and IL-6 predicted resistance to ipilimumab. Baseline elevated IL-2 levels predicted sensitivity to ipilimumab. In patients receiving immune-chemotherapy, a rise in IL-4 levels during treatment was linked to higher OS.	[63]

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
