# Peer review of "Small-Cell Lung Cancer: Is Liquid Biopsy a New Tool Able to Predict the Efficacy of Immunotherapy?"

_biomolecules, 2024, doi:10.3390/biom14040396_

Round 1
Reviewer 1 Report
Comments and Suggestions for Authors
Dear Authors,
I read through this review with great interest. Given the unique characteristics of SCLC, liquid biopsy is a very useful approach to biomarker development. Several reviews have already been published on liquid biopsy for SCLC, but this review summarizes the liquid biopsy tool by CTC, ctDNA, EV, and soluble factor, including biomarker search to evaluate the efficacy of immunotherapy. I would like the authors to re-consider the creation of tables, including Table 1 as shown below, as well as the explanation of abbreviations using footnote. I also believe that the classification of SCLC subtypes by DNA methylation using ctDNA is an important finding that should be addressed in this review, considering that SCLC-I is highly susceptible to immunotherapy. If subtype classification by ctDNA becomes possible, it will have a significant impact on the prognosis prediction and treatment of SCLC.
Major points:
1. I would like to see more detailed information on "Outcome" in Table 1 to see how much the overall survival improved. What kind of CTs (as controls) were used and what was their OS value? Also, what was used as ICI and what is the OS value of CT+ICI? In addition, what OS, CT, PE, and ORR stand for should be listed as the table footnote.
2. Please include the following two articles recently published in Nat Cancer by Chemi et al (3, 1260-1270, 2022) and Cancer Cell by Heeke et al (42, 225–237, 2024) in the section on Circulating tumor DNA (ctDNA). They report that DNA methylation is distinct across SCLC subtypes and longitudinal assessment from liquid biopsies allows tracking of SCLC subtype evolution.
Minor points:
1. Graphical abstract: Please indicate “The Figure was partly generated using Servier Medical Art, provided by Servier, licensed under a Creative Commons Attribution 3.0 unported license”.
2. Line 63: median survival time (MST) or median survival not median survival rate (SM)
3. Line 181-182: I think VE-cadherin is Vascular Endothelial cadherin, not epithelial cadherin.
4. Line 192 & Table 2: I don't know the unit of “8?m”.
5. Line 195: What does PFS stand for?
6. Line 202-203: “7.5 ml” should be corrected to “7.5 ml of blood”.
7. Table 2: “7.5 ml” should be corrected to “7.5 ml of blood”.
8. Line 346: What does CRC stand for?
9. Line 356: What does DDP stand for?
10. Line 362: What do ncRNA, circRNA, and lncRNA stand for?
11. Line 376: What does CAF stand for?
Comments on the Quality of English LanguageI am sure these will be corrected in the English proofreading, but there were typos, duplications, etc. I would like to see them corrected.
Reviewer 2 Report
Comments and Suggestions for Authors
In the review titled “SCLC: Is liquid biopsy a new tool able to predict efficacy of immunotherapy?” Fasano and colleagues summarizes the recent data on biomarkers present in liquid biopsy, focus attention on their role in tumor detection and treatment choice.This biomarkers are circulating tumor DNA (ctDNA), circulating tumor cells (CTCs), extracellular vesicles (EVs) soluble factors and all are involved in carcinogenesis.
Their characterization is important for evaluating of treatment’s effectiveness, related to chemotherapy and chemotherapy associated to immunotherapy.
The manuscript is well written and organized, the references are updated and, in my opinion, it is acceptable for publication in Biomolecules.
